# Cervical Cancer Screening and Associated Barriers among Women in India: A Generalized Structural Equation Modeling Approach

**DOI:** 10.3390/cancers14133076

**Published:** 2022-06-23

**Authors:** Nilima Nilima, Kalaivani Mani, Siddharth Kaushik, Shesh Nath Rai

**Affiliations:** 1Department of Biostatistics, All India Institute of Medical Sciences, New Delhi 110029, India; manikalaivani@aiims.edu; 2Department of Electrical Engineering, Indian Institute of Technology Delhi, New Delhi 110016, India; sidbiochem88@gmail.com; 3Department of Bioinformatics & Biostatistics, School of Public Health & Information Sciences, University of Louisville, Louisville, KY 40202, USA; 4Biostatistics and Bioinformatics Facility, James Graham Brown Cancer Center, University of Louisville, Louisville, KY 40202, USA

**Keywords:** cervical cancer screening, generalized structural equation modeling, mediation effect, NFHS-4, India

## Abstract

**Simple Summary:**

Exploring the barriers and facilitators of cervical cancer screening is essential to reduce the incidence and mortality, particularly in India. There is a paucity of studies presenting the mediation effects of known barriers and facilitators. The study investigates individual-level social barriers, facilitators, and the factors that mediate the relationships between suspected factors and cervical cancer screening. Understanding the mediation analysis and the effect of mediators will help us acquire a profound understanding of the mechanism of action, which will facilitate in devising strategies keeping the most important factor and their mediators in mind.

**Abstract:**

Exploring the barriers and facilitators of cervical cancer screening (CCS) is essential to reduce the incidence and mortality, particularly in low and middle-income countries. The present study investigates the direct, indirect, and total effects of the barriers and facilitators on CCS in India through the generalized structural equation modeling using data from women files of the fourth round of the National Family Health Survey (NFHS-4). Generalized structural equation models were used to quantify the hypothetical pathway via fitting a series of regression equations. Age, body mass index, religion, years of schooling, awareness of sexually transmitted infection, contraception use, lifetime number of sex partners, number of children, and wealth index were shown to have significant direct effects on the CCS. Older women had 1.16 times the odds of getting screened for cervical cancer as compared to their younger counterpart. The odds of CCS among the women in richest wealth quintile is 2.50 times compared to the poorest. Those who are aware of STIs have 1.39 times the odds of getting screened for cervical cancer. Wealth index, years of schooling, and religion have a substantial indirect and total impact on the CCS. The findings will aid in policy formulations for enhancing the CCS in India.

## 1. Introduction

Globally, cervical cancer is one of the leading diseases in women. It is the second-most prevalent cancer among women in developing countries [1]. In 2018, the World Health Organization estimated that 570,000 women were diagnosed with cervical cancer worldwide and about 311,000 women died of the disease [2,3]. In India, about 70% of those diagnosed with cervical cancer are at the advanced stages, with more than 96,000 cases and nearly 60,000 deaths each year [4,5]. Moreover, among developing nations, India accounts for more than 25% of cervical cancer-related deaths [6]. Despite the fact that India had a national program for cancer since 1975, which, in 2010, became a part of a more comprehensive program known as the National Programme for Prevention and Control of Cancer, Diabetes, Cardiovascular Disease and Stroke (NPCDCS) under the flagship of National Health Mission. However, the program lacks the provision of nationwide screening [7]. India has a National Cancer Control Programme (NCCP) to develop strategies for the detection and prevention of cancer and the National Cancer Registry Programme (NCRP) since December 1981 to estimate the burden of cancer in the country [8]. Despite the NCCP existing guidelines, the screening coverage in India is exiguous. Among women in India aged 30–49 years, only 29.8% reported ever having screened for cervical cancer [4]. Moreover, the lack of awareness and limited screening facilities result in the diagnosis of cervical cancer in advanced stages [9]. 

The burden of cervical cancer can be reduced with an inclusive approach to routine screening and treatment. Hence, it is essential to understand the barriers and facilitators that affect cervical cancer screening in India. Apart from the individual level characteristics such as age, body mass index (BMI) [10,11], and education [12], other barriers include social barriers (e.g., religious beliefs, the responsibility of children at home, embarrassment, and lack of independence in making decisions towards health care) [13] and economical and practical barriers [14] (wealth index, distance, lack of awareness of sexually transmitted diseases, and transport challenges) that influence the utilization of screening programs [15]. Women who have been sexually active or had a STI in the past were more likely to get screened for cervical cancer [13]. Women with more sex partners [16] and using oral contraceptives [17] are at a higher risk of cervical cancer. Prevention includes protected sex using condoms and limiting the number of sex partners [16]. Spatial ecological examination of the factors associated with cervical cancer screening (CCS) in India [5] has been previously reported by the authors of this study; however, the presented study is an attempt to go beyond that in search of more concrete individual-level findings. Several systematic reviews and meta-analysis report embarrassment, fear of screening procedure, lack of knowledge and awareness, and transportation and distance issues to be the most common barriers in low- and middle-income countries [7,18,19].

An interesting study on colorectal cancer screening among Hispanics utilized the SEM approach to investigate the direct and indirect pathways through which the cofactors mediate colorectal screening [20]. There remains a paucity of studies presenting the mediation effects of known barriers and facilitators of CCS in India. To the best of our knowledge, there is no study examining the direct, indirect, and total impacts of the social barriers and facilitators of screening in India using nationwide individual-level data. This study examines the pathway between cervical cancer screening and associated factors in the hypothesized conceptual framework through path coefficients. The main aim of this study is to investigate the barriers and facilitators of CCS in India. A structural equation model investigates the direct, indirect, and total impacts of various exogeneous variables on CCS. This study would help public health strategists and officials in designing a practical approach in the management of cervical cancer in India.

## 2. Materials and Methods

### 2.1. Data 

The National Family Health Survey (NFHS) has been conducted to disseminate the information regarding population, health, and nutrition in India. It is coordinated by International Institute for Population Sciences (IIPS), Mumbai, and collected by various agencies across India. A question on cervical cancer screening was first introduced in the NFHS-4 that was conducted in 2015–2016. This study utilizes the data from the nationally representative sample of the NFHS-4. The data was collected through individual household interviews, including four different structured questionnaires (household, biomarkers, man and, woman). A two-stage sampling procedure was adopted for sample selection. The data file consisting of responses to the woman questionnaire (data collected from women in their reproductive ages, 15–49 years) were taken for analysis. More information about the survey and data can be obtained from the link http://rchiips.org/nfhs/nfhs4.shtml (accessed on 10 December 2021). 

### 2.2. Measures 

The primary variable of interest to this study was the status of screening the cervix for cancer (screening; screened = 1/not screened = 0). The concerning question was “women aged 15–49 years who have ever undergone cervix examinations”. The demographic profile of the respondents and barriers of screening were considered as applicable in the conceptual framework. Age (15–34 years/35–49 years); body mass index (BMI) (less than 18.5 ‘underweight’/18.5–25 ‘normal’/25–30 ‘overweight’/more than 30 ‘obese’); years of schooling (continuous); wealth index quintiles (poorest/poorer/middle/richer/richest); religion (Hindu/Muslim/Christian/others); number of children (Children; none/one to two/three to four/five and above); contraception use (condoms/others/no contraception used); barriers in visiting health facility (transport and distance a big problem/only transport a big problem/only distance a big problem/not a big problem); autonomy on health care (respondent herself decides on health care/respondent decides on health care, along with husband/husband, and family decides on health care); lifetime number of sex partners (Sex partners; one/two/more than two); and ever heard of sexually transmitted infection (STI awareness; yes/no).

### 2.3. Analysis

The characteristics included in this study were summarized using frequency (percentages) or median (interquartile range, IQR) as applicable. Associations and comparisons were investigated using a chi-squared test and ranksum test as applicable. Path analysis was developed with an intent to quantify the relationships among multiple variables [21]. Path analysis was very powerful in testing and developing the structural hypothesis with both indirect and direct causal effects. Structural equation modeling is a comprehensive multivariate method to test the directional and nondirectional relationship between variables [22]. A structural equation modeling (SEM) is often drawn as path diagrams to quantify the hypothetical pathway between the endogenous, the exogenous, and the mediating factors, as presented in Figure 1. A common function of path analysis and SEM is mediation, which assumes that a variable can influence an outcome directly and indirectly through another variable known as a mediator. A Structural Equation Model is a combination of two methods, path analysis, and confirmatory factor analysis, which were combined in the early 1970s [23] and become popular in many fields including biomedical research. The mathematical description of SEM can be described using the measurement and structural model presented in Equations (1) and (2), respectively [24].
V_i_ = λ_i_L_i_ + e_i_(1)
where V_i_ is the vector of observed variables, L_i_ is the vector of latent variables, λ_i_ is the vector of parameters, and e_i_ is the vector of measurement errors.
E_i_^*^ = β_i_M_i_ + υ_i_E_i_ + ξ_i_(2)
where β_i_ and υ_i_ are parameter vector; E_i_^*^ and E_i_ are endogenous and exogenous variables, respectively, M_i_ is the mediating variable, and ξ_i_ are the residual terms.

A generalized structural equation modeling (GSEM) is a generalized form of SEM where factor variable notations can be considered while fitting models [25]. Several features of SEM are not available with GSEM including goodness of fit tests. GSEM is fit via a series of simultaneous regression equations [26]. Direct effects are the complete independent effect of an exogenous variable on the endogenous variable. The effect of exogenous variables on endogenous variables via some other endogenous variables is known to be indirect effect, and such a mediating endogenous variable is known as a mediator in the relationship between the two variables of interest. Mediation analysis helps us acquire profound understanding of the mechanism of action of social determinants. This insight creates a new dimension in understanding the etiology of condition and the associated pathways, which can lead to the identification of more effective strategies. The total effect is the summation of direct and indirect effects of exogenous variable on the endogenous variable. The indirect effects, total effects, and the associated tests are not in-built under the post estimation of ‘gsem’ command in Stata and, hence, were decomposed using the non-linear combination command ‘nlcom’. Odds ratios were obtained using ‘estat eform’ command in Stata. A *p* < 0.05 was considered statistically significant throughout. The analysis was run by using Stata v.16 (StataCorp LLC, College Station, TX, USA). 

## 3. Results

The data on CCS was collected from 699,686 women, of which approximately 147,380 (21%) underwent CCS. Among those who underwent CCS, approximately 78,480 (53%) were aged 15–34, 23,127 (83%) were aware of sexually transmitted infections, 109,376 (74%) were Hindus, the majority 109,376 (26%) were in the richest wealth quintile, 74,678 (51%) had one or two children, only 9458 (6%) used condom, 2865 (11%) made independent decisions regarding health care, and 97,774 (66%) did not find distance or transport a big problem in a visiting health facility. The screening status across the various characteristics of interest is described in Table 1. 

The endogenous variable screening had a binomial family and logit link, whereas the mediating variables viz contraception use and number of children were specified to have a multinomial family with logit link, and the mediator STI was specified to have a binomial family with logit link. The resulting model had a Log likelihood = −1,509,984. The GSEM findings, in terms of path coefficients (PC), their 95% confidence intervals, and *p*-values presented in Table 2, revealed the direct effects and the indirect effects of the factors in the screening. The findings revealed that the relationship between CCS and the factors such as years of schooling, religion, and wealth index were mediated by contraception use, STI, and number of children as applicable. For factors that do not have an indirect pathway, as presented in Figure 1, the direct effects are their total effect on screening. For factors with an indirect pathway, the total effect is the sum of direct and indirect effects. The direct effect of certain variables on the mediator is not presented in Figure 1 and can be obtained from the Direct effect on the respective endogenous variables column of Table 2.

The indirect effect on screening in Figure 1c was obtained utilizing the data on the PC from the Direct effect on the respective endogenous variable column of Table 2. The effect, say for, the wealth index, poorer via the number of children (one or two) and wealth index, poorer → the number of children (one or two) → screening is 0.104 × (0.221) = 0.023.

Factors including the age of the respondents, BMI, religion, years of schooling, awareness of sexually transmitted infection, contraception use, lifetime number of sex partners, number of children, and wealth index were shown to have a significant direct effect on the CCS. Autonomy of the respondents (women) in making decision towards their health care was not significantly associated with screening when adjusted for the effect of other factors. Religion and years of schooling are observed to have significant direct effect on contraception use. Religion and wealth index were also noted to have significant direct impacts on the number of children. Wealth index, years of schooling, and religion have a substantial indirect and total impact on the CCS. Years of schooling have shown a significant indirect effect on CCS via the awareness of STI. The adjusted odds ratio (AOR) obtained using a multivariable regression concerning the primary endogenous variable, cervical cancer screening, is presented in Figure 2 for ease in interpretation of the findings presented in Table 2.

Figure 2 shows, that older women (aged 35–49 years) had 1.16 times the odds of getting screened for cervical cancer as compared to their younger counterpart (aged 15–34 years). Overweight and obese women have significantly higher odds (1.13 and 1.21, respectively) of getting screened for cervical cancer as compared to the underweight women. In general, an increase in odds ratio is noted with the increase in BMI among the targeted population. The odds of getting screened for CCS among the women in richest wealth quintile is 2.50 times compared to those in the poorest wealth quintile. The odds of getting screening are noted to be increasing with the increase in the wealth index. Those who are aware of STIs have 1.39 times the odds of getting screened for cervical cancer. The lifetime number of sex partner is also shown to have an influence on the CCS. Women who only ever had one or two sexual partners did not show any significant difference in screening behavior (OR = 0.99, *p* = 0.974). However, women who had more than two sexual partners in their lifetime had 19% lower odds (OR = 0.81, *p* = 0.009) of getting screened for CCS. An interesting finding concerning the years of schooling revealed that, with a year of schooling more, the odds of getting screened for cervical cancer decreases by 1% (*p* < 0.001). 

The respondent’s autonomy on health care and the issues in visiting health care facility were most anticipated factors; however, the findings reveal that these do not have any significant impact on the cervical cancer screening in India. A decrease in odds of getting screened for CCS is noted among women with increasing number of children. Women with more than four children are noted to have no significant difference in the odds of getting screened for CCS as compared to women who have no children at all (OR = 1.01, *p* = 0.821). When compared with those who do not use contraception, the respondents who used contraception other than condom had 1.18 odds (*p* < 0.001), whereas those who used condoms had 1.05 odds (*p* = 0.156) of getting screened for cervical cancer. The lower percentage of respondent using condoms could be an explanation to this and highlights the importance of working towards popularizing this method of contraception.

The odds of getting screened for CCS is 1.18 times more among the women belonging to the Muslim religion as compared to those among the Hindu after adjusting for the other factors in the model. Moreover, no significant difference was noted in the odds of screening between Hindu’s and Christian’s in India (*p* = 0.669). Further investigating the total effects of religion, it was found that there is no significant difference in the odds of getting screened among Hindus and Muslims when mediated by contraception use and number of children. Some of the total effects of religion on CCS were are insignificant when the relationship was moderated by the contraception usage and the various levels of number of children as presented in the *Total effects on screening* column of Table 2. These findings reveal the importance of number of children and contraception use and their effect on the relationships between other exogenous variables and the screening, which needs to be further investigated in depth. 

## 4. Discussion

This study aimed to investigate the barriers and facilitators of cervical cancer screening in India. To facilitate an in-depth investigation of factors and their mediators, if any, a generalized structural equation modeling approach was used on the individual-level nationally representative data from India. The study findings revealed that age of the respondent, BMI, religion, years of schooling, awareness of STI, contraception use, lifetime number of sex partners, number of children, and wealth index had a significant direct impact on the CCS. Previously, an ecological spatial study on CCS in India reported country-wide hot spots and cold spots of screening along with the factors associated [5]. Incorporating the component of geography, the factors related to CCS were the percentage of women—with poor wealth index, not using a modern method of contraception, residing in rural areas, and being aware of STIs [5]—which is in line with the present study findings. To the best of our knowledge, the present study is the first to investigate the mediation effect of factors on the CCS percentage in India. 

A study conducted among 932 women from a state in India reported that educated, younger, and women who used contraception were likely to get screened for CC [27]. The current study findings are in-line in terms of contraception use and age; however, it is different in terms of education status. Education status was used as a proxy for knowledge on CC; however, it was not found to reveal the correct pattern. Upon investigating the effect that years of schooling has on the CCS, mediated via awareness of STIs, we found that years of schooling have a significant direct impact on awareness of STIs. An increase in odds of awareness about STIs was noted with the increase in years of schooling. This could also be indicative of a curriculum that includes content on STIs such as gonorrhea, syphilis, and HIV but lacks content on cervical cancer and the benefits of screening for them. Poor health literacy could better predict the CCS awareness and importance than attending school education [28]. A study from Kenya [29] reports that older women, more than 30, were more likely to get screened for CC, which is in line with the findings of the presented study. A case–control study [30] and other studies [31,32] reported the increased risk of cervical cancer among oral or hormonal contraception users. However, the same study [30] presented the lifetime use of contraceptives to be protective against cervical cancer. Anticipating improved utility of condoms for contraception among the respondents who were aware of the preventive action of condoms and the risk due to hormonal contraception, the present study considered contraception as a factor and found it to have a significant direct impact on the CCS. The current study findings also revealed an increased odds of getting screened among those who use contraception (including oral or hormonal) other than a condom. The present study findings report an association between the lifetime number of sex partners and the CCS. It was noted that respondent with more than two sex partners are less likely to get themselves screened as compared to those with single intimate partner. However, women with up to two sex partners do not seem to show more or less likely to go for CCS. A study evaluating the perceived need of CCS among 219 non-pregnant women in Kenya [29] reported a significant difference in the percentage of women with one (57%) versus more than one (71%) sexual partner, indicating the importance of awareness on how multiple sexual partner can lead to CC and other STIs. The present study ascertains that woman who were aware of STIs were more likely to go for CCS. A study from India reports the lack of awareness and social stigma as the main obstacle for the success of screening programs in India [33]. Women with prior diagnosis of any STI were more likely to go for CCS [34]. Screening program should focus on devising strategies for the recruitment of such high-risk women [34]. Studies report that Muslim women were found to resist the screening practices that they believed threatened their religious values [35,36,37]; however, the present study findings are contradictory and generate a hypothesis that needs further investigation. Christians were found to have good knowledge about CCS [37]. However, the present study findings reveal no significant effect of religion on the CCS when the relationship was moderated by number of child and the contraception usage. It is known that obesity is associated with cervical cancer [11], and hence, the study examined the relationship between BMI and CC screening to find that, as the BMI increases, the odds of getting screened for CCS increases. Women with low incomes or limited access to a health care facility are less likely to be screened for cervical cancer [5,38]. The present study findings reveal an increasing trend in the odds of getting screened as the wealth index improves from poorest to richest. Cost-effective screening services are needed [29]. To reduce the burden of cervical cancer in India, this study recommends a nation-wide subsidized screening program with simultaneous counselling for awareness of the condition and associated risk factors. The number of children is undoubtedly a barrier for women due to increased responsibility at home. Owing to which, number of children was taken as a mediator (barrier) for CCS. A decrease in odds of getting screened for CCS is noted among women with increasing number of children. This is in line with the findings of the study where it was shown that practical barriers are heavily influencing the screening uptake [13]. An interesting article report that the psychological barriers like anxiety, embarrassment, and poor knowledge influence screening but not practical barriers like time, distance, conveyance, etc. [39]. Autonomy and issues in visiting health facility were not found to have a significant impact on the CCS. 

This study provides evidence towards the suspected mediators like number of children, contraceptive usage, and awareness of STIs to be barriers and facilitators, respectively, of CCS among women in India. Considering these barriers and facilitators while devising strategies to improve the screening percentage in India and abroad will help in long run. The role of these barriers and facilitators will help to achieve the goal of cervical cancer elimination through screening in similar settings.

Similar to every study, this study also has a few limitations. A large nationally representative sample of women in their reproductive ages was considered for analysis, which makes the results generalizable only to this age group. The results cannot be generalized for women above 50 years of age. Owing to the large sample size, the bias of the estimate was meagre, and hence, the survey weights were not incorporated for the analysis. The data utilized in this study is collected cross-sectionally, and the findings are presented to facilitate future research in this field. All the limitations associated with the data and collection procedure are also a limitation of this study.

## Figures and Tables

**Figure 1 cancers-14-03076-f001:**
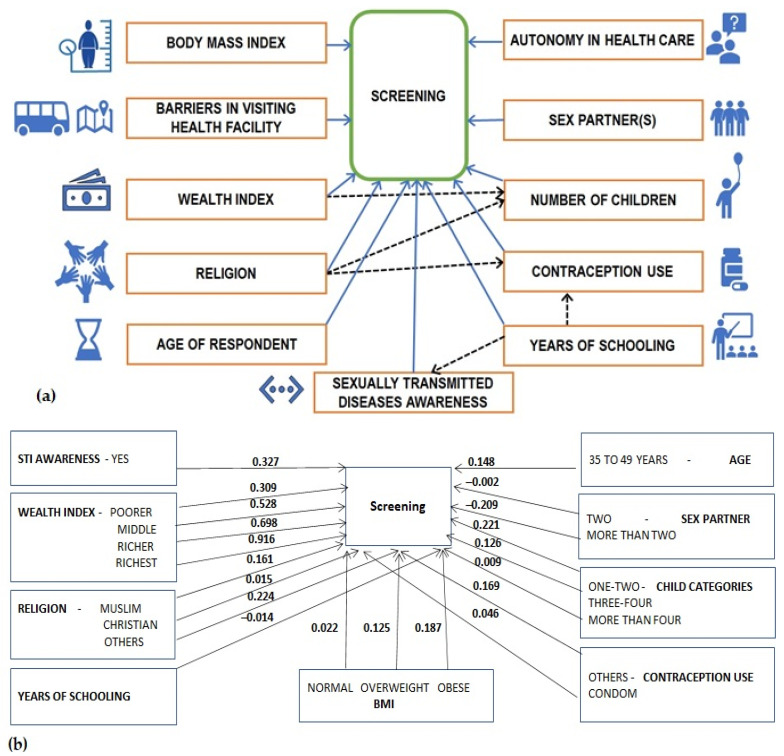
(**a**) Hypothesized conceptual framework; (**b**) significant direct effect; and (**c**) indirect effect mediated via the number of children, STI, and contraception usage in cervical cancer screening in India, NFHS-4, 2015–2016 (Source: Author generated).

**Figure 2 cancers-14-03076-f002:**
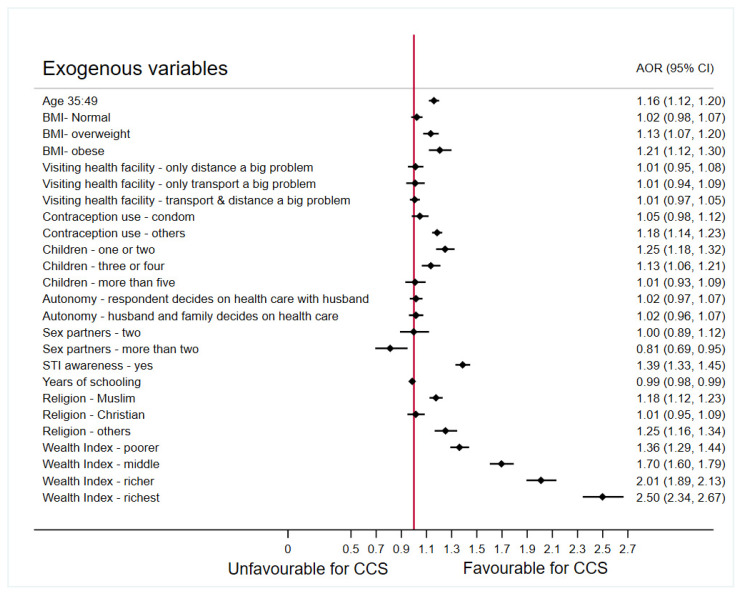
Forest plot presenting the odds ratios adjusted for other factors associated with cervical cancer screening in India, NFHS-4, 2015–2016 (Source: Author generated). Reference categories: Age: 15–34; BMI—underweight; Visiting health facility—not a big problem; Contraception use—none; Children—none; Autonomy—respondent decides on health care alone; Sex partners—one; STI awareness—no; Religion—Hindu; Wealth Index—poorest.

**Table 1 cancers-14-03076-t001:** The cervical cancer screening status across the characteristics of the population, NFHS-4, 2015–2016.

Characteristics	Cervical Cancer Screening		*p*
	Yes	No	
Age—15:34	78,480 (17.06)	381,477 (82.94)	<0.001
Age—35:49	68,900 (28.74)	170,829 (71.26)	
BMI—Underweight	22,438 (14.89)	128,221 (85.11)	<0.001
BMI—Normal	83,587 (20.37)	326,849 (79.63)	
BMI—Overweight	29,397 (30.27)	67,719 (69.73)	
BMI—Obese	9868 (34.09)	19,077 (65.91)	
Visiting health facility—Not a big problem	97,774 (22.16)	343,517 (77.84)	<0.001
Visiting health facility—only distance a big problem	10,596 (21.03)	39,800 (78.97)	
Visiting health facility—only transport a big problem	7468 (20.00)	29,876 (80.00)	
Visiting health facility—transport & distance a big problem	31,542 (18.48)	139,113 (81.52)	
Contraception use—condoms	9458 (32.04)	20,059 (67.96)	<0.001
Contraception use—others	70,756 (30.53)	160,985 (69.47)	
Contraception use—none	67,166 (15.32)	371,262 (84.68)	
Children—none	15,075 (6.76)	207,992(93.24)	<0.001
Children—one or two	74,678 (29.68).	176,906 (70.32)	
Children—three or four	44,925 (27.08)	120,985 (72.92)	
Children—more than four	12,702 (21.48)	46,423 (78.52)	
Autonomy—respondent decides on health care alone	2865 (30.36)	6573 (69.64)	<0.001
Autonomy—respondent decides on health care along with husband	16,976 (30.19)	39,262 (69.81)	
Autonomy—Husband and family decides on health care	6009 (28.43)	15,126 (71.57)	
Sex partners—one	25,482 (29.94)	59,624 (70.06)	<0.001
Sex partners—two	475 (26.60)	1311 (73.40)	
Sex partners—more than two	239(24.24)	747 (75.76)	
STI awareness—no	4809 (17.21)	23,468 (24.86)	<0.001
STI awareness—yes	23,127 (82.79)	70,947 (75.14)	
Years of schooling—median(IQR)	7 (0,10)	8(0,10)	<0.001
Religion—Hindu	109,376 (21.06)	409,905 (78.94)	<0.001
Religion—Muslim	19,247 (20.35)	75,344 (79.65	
Religion—Christian	9454 (18.14)	42,659 (81.86)	
Religion—others	9303 (27.60)	24,398 (72.40)	
Wealth Index—poorest	18,224 (13.68)	115,025 (86.32)	<0.001
Wealth Index—poorer	25,994 (17.39)	123,472 (82.61)	
Wealth Index—middle	30,999 (21.06)	116,169 (78.94)	
Wealth Index—richer	33,826 (24.42)	104,676 (75.58)	
Wealth Index—richest	38,337 (29.20)	92,964 (70.08)	

**Table 2 cancers-14-03076-t002:** The direct, indirect, and total impacts of factors associated with cervical cancer screening in India using generalized structural equation modeling. NFHS-4, 2015–2016.

Endogenous Variables	Exogenous Variables	Direct Effect on Respective Endogenous Variable		Indirect Effect on Screening *		Total Effect on Screening **	
		PC (95% CI)	*p*	PC (95% CI)	*p*	PC (95% CI)	*p*
**Screening (n = 82,533)**	Age—15:34	Ref	-	-	-	-	-
	Age—35:49	0.148 (0.112, 0.183)	<0.001	-	-	-	-
	BMI—Underweight	Ref	-	-	-	-	-
	BMI—Normal	0.022 (−0.022, 0.066)	0.317				
	BMI—Overweight	0.125 (0.071, 0.180)	<0.001				
	BMI—Obese	0.187 (0.112, 0.261)	<0.001				
	Visiting health facility—not a big problem	Ref	-	-	-	-	-
	Visiting health facility—only distance a big problem	0.012 (−0.048, 0.073)	0.695	-	-	-	-
	Visiting health facility—only transport a big problem	0.010 (−0.063, 0.082)	0.795	-	-	-	-
	Visiting health facility—transport and distance a big problem	0.006 (−0.033, 0.046)	0.749	-	-	-	-
	Contraception use—none	Ref	-	-	-	-	-
	Contraception use—condom	0.046 (−0.018, 0.110)	0.156	-	-	-	-
	Contraception use—others	0.169 (0.135, 0.203)	<0.001	-	-	-	-
	Children—none	Ref	-	-	-	-	-
	Children—one or two	0.221 (0.162, 0.280)	<0.001	-	-	-	-
	Children—three or four	0.126 (0.060, 0.192)	<0.001	-	-	-	-
	Children-more than four	0.009 (−0.071, 0.089)	0.821	-	-	-	-
	Autonomy—respondent decides on health care alone	Ref	-	-	-	-	-
	Autonomy—respondent decides on health care along with husband	0.016 (−0.034, 0.066)	0. 521	-	-	-	-
	Autonomy—husband and family decides on health care	0.016 (−0.040, 0.072)	0.576	-	-	-	-
	Sex partners—one	Ref	-	-	-	-	-
	Sex partners—two	−0.002 (−0.117, 0.114)	0.974	-	-	-	-
	Sex partners—more than two	−0.209 (−0.367, −0.052)	0.009	-	-	-	-
	STI awareness—no	Ref	-	-	-	-	-
	STI awareness—yes	0.327 (0.285, 0.370)	<0.001	-	-	-	-
	Years of schooling	−0.014 (−0.017, −0. 010)	<0.001	-	-	-	-
	Religion—Hindu	Ref	-	-	-	-	-
	Religion—Muslim	0.161 (0.117, 0.206)	<0.001	-	-	-	-
	Religion—Christian	0.015 (−0.053, 0.083)	0.669	-	-	-	-
	Religion—others	0.224 (0.015, 0.295)	<0.001	-	-	-	-
	Wealth Index—poorest	Ref	-	-	-	-	-
	Wealth Index—poorer	0.309 (0.254, 0.364)	<0.001	-	-	-	-
	Wealth Index—middle	0.528 (0.472, 0.584)	<0.001	-	-	-	-
	Wealth Index—richer	0.698 (0.639, 0.757)	<0.001	-	-	-	-
	Wealth Index—richest	0.916 (0.851, 0.981)	<0.001	-	-	-	-
**Contraception Use—No usage versus condom (n = 699,686)**	Religion—Hindu	Ref	-	Ref	-	Ref	-
	Religion—Muslim	0.197 (0.165, 0.229)	<0.001	0.009 (−0.004, 0.022)	0.158	Via no.of child level 1	
						0.056 (−0.001, 0.112)	0.051
						Via no.of child level 2	
						0.149 (0.102, 0.198)	<0.001
						Via no.of child level 3	
						0.175 (0.117, 0.233)	<0.001
	Religion—Christian	−1.719 (−1.807, −1.631)	<0.001	−0.080 (−0.190, 0.030)	0.156	Via no.of child level 1	
						−0.153 (−0.279, −0.028)	0.017
						Via no.of child level 2	
						−0.089 (−0.214, 0.036)	0.162
						Via no.of child level 3	
						−0.062 (−0.191, 0.066)	0.339
	Religion—others	0.506 (0.461, 0.551)	<0.001	0.023 (−0.009, 0.060)	0.156	Via no.of child level 1	
						0.240 (0.162, 0.318)	<0.001
						Via no.of child level 2	
						0.237 (0.159, 0.316)	<0.001
						Via no.of child level 3	
						0.245 (0.162, 0.327)	<0.001
	Years of schooling	0.065 (0.062, 0.067)	<0.001	0.003 (−0.001, 0.007)	0.156	Via STI 0.053 (0.045, 0.063)	<0.001
**Contraception Use—No usage versus Others (n = 699,686)**	Religion—Hindu	Ref	-	Ref	-	Ref	-
	Religion—Muslim	−0.636 (−0.653, −0.620)	<0.001	−0.108 (−0.130, −0.086)	<0.001	Via no.of child level 1	
						−0.061 (−0.115, −0.007)	0.027
						Via no.of child level 2	
						0.033 (−0.015, −0.081)	0.179
						Via no.of child level 3 0.058	
						(−0.005, 0.122)	0.070
	Religion—Christian	−0.773 (−0.796, −0.751)	<0.001	−0.131 (−0.158, −0.104)	<0.001	Via no.of child level 1	
						−0.204 (−0.277, −0.132)	<0.001
						Via no.of child level 2	
						−0.140 (−0.211, −0.069)	<0.001
						Via no.of child level 3	
						−0.113 (−0.187, −0.040)	0.003
	Religion—others	0.029 (0.005, 0.054)	0.017	0.005 (0.001, 0.009)	0.021	Via no.of child level 1	
						0.222 (0.150, 0.293)	<0.001
						Via no.of child level 2	
						0.219 (0.147, 0.291)	<0.001
						Via no.of child level 3	
						0.226 (0.151, 0.301)	<0.001
	Years of schooling	−0.065 (−0.067, −0.062)	<0.001	−0.015 (−0.019, −0.012)	<0.001	Via STI 0.035 (0.026, 0.045)	<0.001
**Children (One–two) (n = 699,686)**	Wealth Index—poorest	Ref	-	Ref	-	Ref	-
	Wealth Index—poorer	0.104 (0.084, 0.123)	<0.001	0.023 (0.015, 0.030)	<0.001	0.332 (0.276, 0.387)	<0.001
	Wealth Index—middle	0.190 (0.171, 0.209)	<0.001	0.042 (0.030, 0.054)	<0.001	0.570 (0.513, 0.627)	<0.001
	Wealth Index—richer	0.302 (0.283, 0.321)	<0.001	0.068 (0.048, 0.085)	<0.001	0.765 (0.703, 0.827)	<0.001
	Wealth Index—richest	0.383 (0.365, 0.403)	<0.001	0.085 (0.062, 0.108)	<0.001	1.001 (0.932, 1.069)	<0.001
	Religion—Hindu	Ref	-	Ref	-	Ref	-
	Religion—Muslim	−0.518 (−0.535, −0.500)	<0.001	−0.115 (−0.145, −0.084)	<0.001	-	-
	Religion—Christian	−0.399 (−0.421, −0.377)	<0.001	−0.088 (−0.112, −0.064)	<0.001	-	-
	Religion—others	−0.032 (−0.058, −0.005)	0.017	−0.007 (−0.013, −0.001)	0.024	-	-
**Children (three–four)(n = 699,686)**	Wealth Index—poorest	Ref	-	Ref	-	Ref	-
	Wealth Index—poorer	−0.134 (−0.154, −0.114)	<0.001	−0.017 (−0.026, −0.008)	<0.001	0.292 (0.236, 0.348)	<0.001
	Wealth Index—middle	−0.287 (−0.307, −0.267)	<0.001	−0.036 (−0.055, −0.017)	<0.001	0.492 (0.433, 0.551)	<0.001
	Wealth Index—richer	−0.440 (−0.460, −0.419)	<0.001	−0.055 (−0.084, −0.026)	<0.001	0.643 (0.577, 0.708)	<0.001
	Wealth Index—richest	−0.692 (−0.714, −0.671)	<0.001	−0.087 (−0.133, −0.042)	<0.001	0.829 (0.750, 0.907)	<0.001
	Religion—Hindu	Ref	-	Ref	-	Ref	-
	Religion—Muslim	−0.165 (−0.183, −0.146)	<0.001	−0.021 (−0.032, −0.0097)	<0.001	-	-
	Religion—Christian	−0.192 (−0.216, −0.168)	<0.001	−0.024 (−0.037, −0.011)	<0.001	-	-
	Religion—others	−0.079 (−0.110, −0.048)	<0.001	−0.010 (−0.016, −0.003)	0.003	-	-
**Children (more than four) (n = 699,686)**	Wealth Index—poorest	Ref	-	Ref	-	Ref	-
	Wealth Index—poorer	−0.518 (−0.543, −0.494)	<0.001	−0.005 (−0.046, 0.037)	0.821	0.304 (0.236, 0.372)	<0.001
	Wealth Index—middle	−0.994 (−1.020, −0.967)	<0.001	−0.009 (−0.089, 0.070)	0.821	0.519 (0.425, 0.613)	<0.001
	Wealth Index—richer	−1.470 (−1.501, −1.439)	<0.001	−0.014 (−0.131, 0.104)	0.821	0.684 (0.556, 0.812)	<0.001
	Wealth Index—richest	−2.098 (−2.137, −2.058)	<0.001	−0.019 (−0.187, 0.149)	0.821	0.896 (0.721, 1.072)	<0.001
	Religion—Hindu	Ref	-	Ref	-	Ref	-
	Religion—Muslim	0.507 (0.484, 0.531)	<0.001	0.005 (−0.036, 0.045)	0.821	-	-
	Religion—Christian	0.262 (0.229, 0.294)	<0.001	0.002 (−0.019, 0.023)	0.821	-	-
	Religion—others	−0.306 (−0.361, −0.250)	<0.001	−0.003 (−0.027, 0.022)	0.821	-	-
STI	Schooling	0.197 (0.194, 0.200)	<0.001	0.064 (0.056, 0.073)	<0.001	-	-

* Via the respective endogenous variable level (The rows in which values are reported). ** Via one or more endogenous variable levels (The row in which values are reported along with the “via”, if specified).

## Data Availability

The data are freely available and accessible from http://rchiips.org/nfhs/nfhs4.shtml.

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
