# Peer review of "Cervical Cancer Screening and Associated Barriers among Women in India: A Generalized Structural Equation Modeling Approach"

_cancers, 2022, doi:10.3390/cancers14133076_

Round 1

Reviewer 1 Report

This study is interesting and well explained, some precisions and better explanations would be desirable, I added my comments directly in the PDF.

Reviewer 2 Report

The authors attempted to identify factors and mediators for cervical cancer screening among women in India using the NFHS dataset for cycles 2015-2016. This is an interesting and well-written study and may help facilitators to develop a policy for enhancing screening uptake.  Several modifications and clarifications may be given to improve the quality of the manuscript:

  1. The authors have used a generalized structural equation modeling approach (GSEM) for detecting mediators for screening uptake using the NSHS database. The NSHS uses a two-stage sampling procedure and thus survey weighting should be incorporated into the analyses. Otherwise, the estimation would be biased.
  2. In cross-sectional data, it is hard to declare mediators in the conceptual framework. It is unclear how the authors have chosen the appropriate mediators for the analyses. The selected mediators could be probable effect modifiers or cofounders rather than true mediators. In addition, there could be some additional mediators such as STI awareness may be a mediator between years of schooling and CCS, barriers to visiting health facilities may be a mediator between wealth and CCS. Given the cross-sectional nature of the study, I would suggest exploring alternative models by switching the mediators based on theoretical framework and health belief models and the model producing a better fit to the data as measured by fit indices should be used to finalize the GSEM model. I would suggest citing and refer the following recent manuscript:
    1. Shokar, N. K., Salinas, J., & Dwivedi, A. (2022). Mediators of screening uptake in a colorectal cancer screening intervention among Hispanics. BMC cancer22(1), 1-13.
  3. Please define the study eligibility criteria. Typically, age at cervical cancer screening starts at age 25. However, the authors have included participants from ages 15 to 49. What about ages 50 to 65?
  4. The primary outcome was defined as “the cervix for cancer screening (screened=0/not screened=1)” but in table1, the authors wrote CCS screened- yes and no. Is yes represent 1 and no represent 0?  The title of tables and figures reflect screening/CCS but results are presented with not-screened.
  5. The interpretation includes two negative terms such as “the odds of not getting screened for CCS is lesser”. I would suggest interpreting in terms of the odds of CCS being lower or higher to improve readability. For example, if the OR=0.85 then the authors may simply say that the odds of CCS is 15% lower in age>35 compared to age<=35 or the authors may say that the odds of CCS is 18% higher in age>35 compared to age<=35.
  6. In the forest plot figure 2, it would be better to provide favorable for screening and unfavorable for screening on the x-axis. In addition, referent categories should be specified in the exogenous variables.
  7. Some categories of variables had less than 1-2% data, and contraception use-condoms had less than 5% data. For example, sex partners-three; more than 3. Why wealth was categorized into categories but not the years of education? Some critical variables such as location (rural/ urban), marital status, and obesity status may be included in the analyses.
  8. Table 1 percentages may be presented row-wise instead of column-wise.
  9. Please specify the limitations and strengths of the study.

Reviewer 3 Report

In their study, the authors studied the mediating factors leading to poor cervical cancer screening in India. They outline the direct and indirect factors that mediate the relationship using structural equation modeling. This is a very important study and well done; however, I think it could be improved to improve the clarity of the methodology and clarity of the results. Additionally, the entire manuscript should be proofread multiple times to address vocabulary/grammar errors throughout the work.

Introduction:

  • There are multiple errors in the writing where words have no spaces between them. Please address this.
  • Last paragraph of the introduction, second to last sentence “A structural equation model…”- this would seem to fit better in the methods.

Methods:

  • I really enjoyed the structural equation modeling approach to identify the mediators in the pathway; however, I would like to see more discussion as to why these particular variables were chosen
    • Similarly, I would like to see a discussion as to why the authors considered them as mediators in the different pathways (i.e in Figure 1, why are the pathways drawn as they are). This should either be based on quantitative methods assessing these variables as mediators or based on published data that would suggest these as mediators in a pathway.
  • Were other regression models considered in the gsem model (i.e. Poisson, negative binomial, etc). Authors should include this consideration and possibly fit models showing that logit is better.

Results:

  • The results section can use a lot of work. It was confusing as the reader understanding how variables were related and what the findings were.
    • Firstly, there should not be double negatives as this is very very confusing. For example “women aged 35-49 had 15% lesser odds of not going for screening” is a double negative and very confusing. Does this meaning they had 15% higher odds of screening? Or 15% lower odds of screening completion? There are other instances of this throughout the results that need to be addressed.
    • Please report OR to 2 decimal places.

Discussion:

  • While I think the discussion is good, it needs to be reread and edited multiple times as there are many grammar mistakes throughout, in fact too many to list. Please ensure that this is proofread prior to resubmitting
  • Finally, I would like to see a discussion as to how the authors would address the mediating factors that they identified. What can lawmakers/policymakers/advocates do with this information in India?

Round 2

Reviewer 1 Report

congratulations to the authors for this new version of the article which seems to me clearer for the reader

Reviewer 2 Report

You may want to acknowledge the limitations of why weight could not be incorporated into the analysis.